# Biological Activity of Selenium and Its Impact on Human Health

**DOI:** 10.3390/ijms24032633

**Published:** 2023-01-30

**Authors:** Giuseppe Genchi, Graziantonio Lauria, Alessia Catalano, Maria Stefania Sinicropi, Alessia Carocci

**Affiliations:** 1Dipartimento di Farmacia e Scienze della Salute e della Nutrizione, Università della Calabria, Arcavacata di Rende, 87036 Cosenza, Italy; 2Dipartimento di Farmacia-Scienze del Farmaco, Università degli Studi di Bari “A. Moro”, 70125 Bari, Italy

**Keywords:** selenium, selenoproteins, selenium deficiency, human diseases, epigenetics, phytoremediation, rhizofiltration

## Abstract

Selenium (Se) is a naturally occurring metalloid element essential to human and animal health in trace amounts but it is harmful in excess. Se plays a substantial role in the functioning of the human organism. It is incorporated into selenoproteins, thus supporting antioxidant defense systems. Selenoproteins participate in the metabolism of thyroid hormones, control reproductive functions and exert neuroprotective effects. Among the elements, Se has one of the narrowest ranges between dietary deficiency and toxic levels. Its level of toxicity may depend on chemical form, as inorganic and organic species have distinct biological properties. Over the last decades, optimization of population Se intake for the prevention of diseases related to Se deficiency or excess has been recognized as a pressing issue in modern healthcare worldwide. Low selenium status has been associated with an increased risk of mortality, poor immune function, cognitive decline, and thyroid dysfunction. On the other hand, Se concentrations slightly above its nutritional levels have been shown to have adverse effects on a broad spectrum of neurological functions and to increase the risk of type-2 diabetes. Comprehension of the selenium biochemical pathways under normal physiological conditions is therefore an important issue to elucidate its effect on human diseases. This review gives an overview of the role of Se in human health highlighting the effects of its deficiency and excess in the body. The biological activity of Se, mainly performed through selenoproteins, and its epigenetic effect is discussed. Moreover, a brief overview of selenium phytoremediation and rhizofiltration approaches is reported.

## 1. Introduction

Selenium (Se) is a trace mineral, ubiquitously occurring in the environment that is of fundamental importance to human health. Initially regarded as a toxic element, its role as an essential element in the body was not established until 150 years after its discovery in 1957, when Klaus Schwartz and Calvin Foltz recognized it as a substance responsible for preventing lesions in the liver, blood vessels, and muscles in rats and chickens [1,2]. Since then, its impact on the human body and the mechanisms involved have been thoroughly investigated, being the link between Se deficiency, as well as its excess, and disease occurrence an important aspect. Despite its very low level in humans, Se plays an important and unique role among the trace essential elements being the only one for which incorporation into proteins is genetically encoded, as the constitutive part of the 21st amino acid, selenocysteine. In the form of selenocysteine, it is present in the active centers of Se-dependent enzyme (glutathione peroxidases, thioredoxin reductases, and iodothyronine deiodinases). Proteins having at least one selenocysteine residue in their structure are called selenoproteins that perform many important physiological roles, their main function being the maintenance of the redox balance in cells [3]. Data from mainly murine and cell-based, but also some human studies, show that Se supplementation modifies epigenetic marks. Conceivably, this occurs through DNA methyltransferase inhibition by Se and its interaction with one-carbon metabolism [4]. Adequate levels of Se are functionally important for several aspects of human biology including male reproductive biology, endocrine system, muscle function, central nervous and cardiovascular systems, and immunity [5]. Prolonged Se deficiency in human organisms leads to serious diseases, since it adversely affects the cardiovascular system functioning, and can be a direct cause of myocardial infarction. The best-known endemic diseases caused by Se deficiency, otherwise known as “geochemical diseases”, are Keshan and Kashin-Beck [6]. Low levels of Se in the body are also responsible for impaired fetal development, infertility in men [7], and increased risk of suffering from asthma because of the reduction in antioxidant defense and the decrease in glutathione peroxidase (Gpx) activity [8]. There is also evidence that Se deficiency weakens the immune system [9] and affects the proper functioning of the nervous system [10]. The main natural source of Se is food where it can exist in inorganic (such as selenates or selenites) and organic forms (selenomethionine, SeMet, and selenocysteine, SeC) [11]. Se deficiency, which affects about one billion people in the world, is due to its insufficient consumption. This factor mainly depends on the geographical area and correlates with the low content of this microelement in the soil, being climate-soil interaction as the main controlling factor [12]. According to World Health Organization (WHO) standards, the recommended dose of Se for adults is 55 μg/day, while the maximum tolerable adult intake without side effects is set at 400 μg/day [13]. Recently the reference values for Se intake have been revised, using the saturation of selenoprotein P (SePP) in plasma as a criterion for the derivation of such values [14]. The integration of selenium-rich food into the diet represents a meaningful measure to avoid deficiency; however, supplemental intake beyond the amounts needed for the full expression of selenoproteins potentially increases health risks therefore, it is not recommended. Excessive supplementation or a diet rich in products with a high content of Se may result in poisoning. In Venezuela, the consumption of the fruit of the species *Lecythis ollaria* which accumulates huge amounts of Se (7–12 g Se/kg of dry matter) caused acute Se poisoning with hair loss, diarrhea, and emesis [15]. Se toxicity depends on its chemical form, besides ingested dose, interactions with other dietary components, and physiological condition of the body. The inorganic forms of Se exhibit higher toxicity than the organic ones. Inorganic Se has a prooxidant effect on thiols, producing free oxygen radicals (ROS), while organic forms are excreted more easily [16]. Acute Se poisoning is difficult to diagnose since its symptoms are rather non-specific. These include hypotension, tachycardia, and neurological disorders such as tremor and muscle contractions [17]. Chronic Se toxicity, otherwise known as selenosis, is characterized by hair loss, changes, and fragility of fingernails, skin rash, joint pain, tooth decay, and a specific garlic odor in the exhaled breath due to the presence of the volatile compound dimethyl selenide [18]. Furthermore, a recent study showed the link between increased Se intake and the risk of type 2 diabetes mellitus [19]. Plants can absorb, assimilate, and accumulate Se in leaves and roots. The capability of plants to take up a substantial amount of Se is now being utilized to remove excess Se from contaminated areas in a process known as ‘phytoremediation’. Phytoremediation of Se-contaminated soils can be a non-polluting and cost-effective way to remove Se that might otherwise be leached out of the soil by excessive irrigation or rainwater to contaminate groundwater, surface waters, or drainage waters [20]. The efficiency of phytoremediation may be greatly increased through the application of recent technological advances in plant breeding, and genetic engineering, and by manipulation of agronomic practices. With this review, we aim to underline the role of Se in the modulation of numerous biological effects, mainly performed through selenoproteins, along with its epigenetic effect. Furthermore, the most recent line of evidence concerning the human health effects of selenium is summarized highlighting the effects of its deficiency and excess in the body which represent the key issue currently at the forefront of this research.

## 2. Selenium Chemistry

Se is a chemical element with atomic number 34 and an atomic weight of 78.96 (Table 1). Se is a nonmetal with properties between sulfur and tellurium (elements above and below in the periodic table) and belongs to the main Group 16 (IV A). Se is a constituent of rare minerals, including crookesite [Cu_7_(Tl,Ag)Se_4_], berzelianite (Cu_2_Se), tiemannite (HgSe), antimonselite (Sb_2_Se_3_) [21,22], and it is obtained during the electrolytic refining of copper. Se was discovered in 1817 by J.J. Berzelius, who was working with sulfuric acid. Berzelius was intrigued by a red sediment collected at the bottom of the container in which the sulfuric acid was prepared. Se is present in different allotropes (gray, red and black) that interconvert with temperature changes and on the rate of temperature change. Prepared in a chemical reaction, selenium is present as an amorphous brick-red powder; rapidly melted, it forms black vitreous beads. Black selenium is a brittle and lustrous solid. The structure of the black form of selenium is irregular and consists of polymeric rings with more than 1000 atoms per ring. Upon heating, it softens and about at 180 °C converts to gray. Gray selenium is the most stable and dense form with a chiral hexagonal crystal lattice consisting of a helical polymeric chain and behaves as a semiconductor with an appreciable photoconductivity. Selenium resists oxidation by air and it is not attacked by oxidizing acids, while with strong reducing agents it forms polyselenides (Se_n_)^2–^. Natural selenium has five stable isotopes (^74^Se, ^76^Se, ^77^Se, ^78^Se, and ^80^Se); of which the most abundant isotopes are ^80^Se and ^78^Se (50% and 23.5%. natural abundance, respectively). Sixteen radioactive isotopes have been synthesized by irradiating selenium nuclei with neutrons. ^79^Se occurs in very little quantities in uranium ores as a product of nuclear fission; it emits beta particles forming ^79^Br with a half-life of 3.27 × 10^5^ years. ^82^Se has a half-life of 9.2 × 10^19^ years and emits double beta particles forming ^82^Kr. It is an extremely rare element; in fact, it is the 59^th^ most common in the Earth’s crust. The average selenium content in the Earth’s crust is about 50 μg/kg, and its concentration in different geographic regions (China, Japan, Russia, Canada, USA) varies from 10 to 2000 μg/kg [15]. In Europe, the major producers of selenium are Finland, Belgium, Germany and England. Se is present in the atmosphere as a component of the volcanic activity and burning of fossil fuel [23]. Se and its inorganic compounds can be found in several geographical sites, such as the lithosphere, hydrosphere, atmosphere, and biosphere. Its chemical form depends on pH and redox properties of the soil, absorption and deposition effects, and biological processes in the presence of microorganisms [24]. Se is present in all living organisms (humans, animals, and plants) as inorganic and organic compounds. The inorganic forms are elementary selenide (Se^2–^), selenite (SeO_3_^2–^), and selenate (SeO_4_^2–^); while the main organic compounds are selenomethionine, selenocysteine, methylselenocysteine, selenocystathionine (Table 2) and proteins containing these amino acids. Plants can absorb and transform inorganic and organic Se forms. The amounts of Se in plants are determined by the type of soil and its pH and salinity, levels in the soil, temperature, and amount of precipitation [25,26]. The mechanisms of intestinal absorption of Se are different in relation to its chemical form. Selenite is absorbed by simple diffusion, while selenate is imported by a cotransport sodium/selenate, OH^–^ antiporter. Se-containing aminoacids are absorbed by Na-dependent aminoacid transport [27]. Some elements (sulfur, lead, arsenic, calcium, and iron) decrease the uptake of selenium. Indeed, Fe^3+^ precipitates Se to an inassimilable complex form; sulfur decreases the absorption of selenium by steric competitiveness.

## 3. Selenium Uses and Applications

In nature, Se is found in sulfide ores of copper, lead, nickel, gold, and silver. Se is commonly produced from selenide in many sulfide ores of copper, nickel, and lead. It is also obtained as a byproduct from the anode mud of copper electrolytic refineries in the proportion of 5 to 25%. Se can be extracted from sludge with the process of roasting with sod crystal or with sulfuric acid treatment. The largest commercial use of selenium (about 50%) is for the production of glass. Selenite (SeO_3_^2–^) and selenate (SeO_4_^2–^) sodium compounds are used in the production of glasses to give them a pink or red color and to hide the green tint that arises in the presence of iron impurities [28]. Se is used as a pigment also for ceramics and plastics. Furthermore, Se is used in photocells, solar cells, and photocopiers. The lithium-selenium battery is in electric vehicles one of the most promising systems for energy storage. It is an alternative to lithium-sulfur batteries with a good advantage of high electrical conductivity and better electrochemical performance [29]. It has been found that the use of Se as fertilizer nullifies the accumulation of lead and cadmium in lettuce. In low doses, Se has shown a beneficial effect on plant resistance to various environmental stress such as drought, UVB, soil salinity, and hot temperatures [30]. Se can also be used to reduce the transmission of sunlight in glass giving it a bronze tint. Se converts light to electricity (photovoltaic action) and has a photoconductive action (electrical resistance decreases with an increase in illumination). Organoselenium compounds are used as a catalyst in some chemical reactions such as selenocyclization, oxyselenenylation, oxidation, and reduction [31]. As a catalyst, it has the advantages of mild conditions, low cost, and it can be recycled and used more and more times. In metallurgy, Se is used to prepare alloys to provide resistance to corrosion and oxidation of metals. It is used for the vulcanization of rubber, for the preparation of pharmaceutical products, and for veterinary uses. Se is involved in the preparation of dietary supplements, and in the treatment of seborrheic dermatitis and dandruff; ^75^Se is applied and used as a gamma source in industrial radiography of welds for steel thickness over 5 mm [32]. ^75^Se is utilized also in biochemistry to follow the metabolism of selenocysteine and selenoproteins [33].

## 4. Selenium in Human Health and Diseases

Humans and animals require Se in trace quantities, being a component of the amino acid selenocysteine present in the active site of the enzymes. At high concentrations, it becomes toxic since it replaces sulfur in enzymes. Se participates in many metabolic processes in the human and animal bodies. In the immune system, Se stimulates the activity of immune cells such as helper T, cytotoxic T, and Natural Killer (NK) [34]. In the human body, selenium deficiency can cause or induce diseases such as Keshan and Kashin-Beck diseases [35]. Furthermore, Se deficiency is associated with muscle necrosis, hypothyroidism, cardio-cerebrovascular disease, male infertility, increased incidence of various cancer, and an improved immune system [21,36]. Se deficiency has supposed to be linked also to infections such as Coronavirus disease 2019 (COVID-19) and acquired immune deficiency syndrome (AIDS) [37]. The Institute of Medicine (USA 2000) has proposed for adult humans a recommended dietary allowance of 55–75 μg/day. Unfortunately, the diet of about one billion people lacks sufficient Se for their good health [35,38]. A large part of dietary selenium in human beings derives directly and indirectly from plants and vegetables (The occurrence of Se in food products has been reported in Table 3). Se levels in soil generally reflect its presence in food; consequently, the lack of this element in human consumption is usually attributed to crop production in geographical areas with low selenium content, and to the amount that edible plants can extract from this soil [21,35,39]. On the other hand, excessive dietary selenium intake may impose risks and damage on human health. The symptoms of selenosis in humans include garlicky breath, dermatitis, hair and fingernail loss, acute respiratory distress, myocardial infarction, and renal failure [21,35,36,40]. The loss of human hair and nails was observed with a very high dietary selenium intake of 2000 μg/day in people of Enshi (Hubei, China) [41,42]. Se, as an antioxidant agent, shows great potential for redox regulation and the maintenance of cellular homeostasis and metabolism [43]. In recent years, the primary role of an excess of reactive oxygen species in the complex pathogenesis of metabolic diseases has been unveiled [44]. Recently, several studies have reported that serum Se status is related to the risk of metabolic diseases therefore, Se supplementation has supposed to be a promising approach in patients with low Se levels. Moreover, the crucial role of Se in chronic metabolic diseases, including cardiovascular disease [45], type 2 diabetes mellitus (T2DM) [46], and nonalcoholic fatty liver disease (NAFLD) [47] has been reported. Several prospective investigations have declared the association between Se level and coronary heart disease (CHD) risk and outcomes. These studies indicate an inverse correlation between Se plasma levels and CHD risk and suggest that moderate Se supplementation may be beneficial in preventing CHD risk. However, the preliminary results of studies conducted on animal models may not truly reflect the effectiveness of Se supplementation in preventing or treating CHD [48]. Se has been reported to play a significant role in glucose and lipid metabolism. Recent studies indicate that high levels of Se are associated with insulin resistance and dyslipidemia. A large cross-sectional study of 8198 rural Chinese reported that serum Se levels were positively correlated with total cholesterol, triglyceride, high-density lipoprotein, and low-density lipoprotein levels, and elevated serum Se levels were related to an increased risk of dyslipidemia [49]. On the other hand, a cross-sectional analysis of 4339 participants found that Se levels were positively associated with insulin resistance and blood glucose; indeed, a 10 μg/L increase in Se was associated with a 1.5% increase in insulin [50]. Based on the role of Se in glucose metabolism and antioxidant defense, Se may be involved in the pathogenesis of type 2 diabetes mellitus (T2DM). Several studies have reached a consensus that high Se exposure is a risk factor for T2DM however, the relationship between dose and effect in observational studies, and its specific role and mechanism have yet to be fully elucidated [50,51,52,53]. NAFLD, the most common cause of liver disease worldwide, is a complex disease that is modulated by numerous mechanisms, including metabolic factors. Some injurious processes, such as oxidative stress contribute to liver damage. Due to the role of Se in lipid metabolism and antioxidant defense, extensive evidence has suggested that Se may contribute to the development of NAFLD. Several animal studies have indicated that Se exposure could induce increased serum liver enzyme levels, the activation of Kupffer cells, and higher hepatic insulin resistance and triglyceride levels in animals, suggesting that Se exposure may be associated with the development of NAFLD [54]. In contrast, Reja et al. reported an inverse relationship between serum Se levels and the risk of advanced liver fibrosis, indicating that Se may be beneficial for the prevention of liver fibrosis in the development of NAFLD [55]. Therefore, whether Se levels are positively or negatively associated with NAFLD risk in humans remains unclear and research opinion regarding the protective action of Se in NAFLD remains inconsistent. Several recent manuscripts reviewed the antioxidant role of nutritional supplementation of selenium in the management of major chronic metabolic disorders including hyperlipidemia and hyperglycemia, highlighting the complex physiological role of Se [43,56,57]. Although the role of Se in metabolic diseases remains unclear, the antioxidant activity of Se in the pathogenesis of diseases cannot be ignored, and the molecular mechanisms underlying these paradoxical effects need to be further explored. Therefore, the relationship between Se status and various health outcomes, especially in metabolic diseases, requires close attention.

## 5. Biosynthesis of Selenocysteine and Selenoproteins

Se is an essential trace element to human health whose beneficial effects are mostly due to its incorporation in the form of selenocysteine (SeC) into a group of proteins called selenoproteins. SeC (considered as a 21st amino acid) is a functional analog of cysteine in which the sulfur atom is replaced by a selenium one. Selenoprotein biosynthesis is a complex process. Since there is no free SeC in the body, its synthesis occurs on specific tRNA (selenocysteinyl tRNA^SeRSeC^) with the UCA anticodon complementary to the UGA stop codon as represented in Figure 1 [27,68]. SeC is incorporated into proteins via tRNA^SerSeC^ that decodes the UGA codon as selenocysteine instead of a stop codon [69,70,71]. In eukaryotes, the synthesis of SeC begins so that seryl-tRNA synthetase charges tRNA^SerSeC^ with serine in the presence of ATP. Subsequently, the hydroxyl mojety of serine is phosphorylated by *O*-phosphoseryl-tRNA kinase (PSTK) in the presence of ATP; finally, *O*-phosphoseryl tRNA^SerSeC^ is substituted by Se atom in the presence of seleno phosphate and the enzymes seleno phosphate synthetase 2 (SPS2) and selenocysteinyl-tRNA synthase (SepSerS) (see Figure 1) [70,72]. The selenocystenyl tRNA^SerSeC^ read UGA codon and is used for the integration of SeC into the amino acid sequence forming selenoprotein. SeC is encoded by the UGA codon, one of the three stop codons necessary for ending the polypeptide chain. The UGA codon encodes selenoprotein only in the presence of SeC insertion sequence (SECIS) element, and protein factors including the SECIS binding protein 2 (SBP2) [73,74,75], which is a nuclear protein, although it functions as a SECIS binding protein 2 in the cytoplasm [76].

## 6. Selenoproteins and Selenoenzymes

Se is incorporated as SeC into proteins and 25 selenoproteins and selenoenzymes have been identified in humans. These proteins include enzymes such as five glutathione peroxidases (GPX), three iodothyronine deiodinases, three thioredoxin reductases, selenophosphate synthetase 2, methionine sulfoxide reductase B1 and selenoproteins F, H, I, K, M, N, O, P, S, T, V, and W [77,78]. Glutathione peroxidase, iodothyronine deiodinase, and thioredoxin reductase are the best-known and the more important selenoenzymes. Among the selenoproteins, glutathione peroxidases are a family of mammalian antioxidant enzymes. Their most important functions are to reduce the hydrogen peroxide to water and lipid hydroperoxides to corresponding ethers in the intracellular and extracellular compartments. These enzymes have glutathione (GSH) as a cofactor, and, in turn, GS-SG (the oxidized form of GSH) is reduced by glutathione reductase [79]. In humans, five GPXs (GPX1-GPX4, and GPX6) contain selenocysteine in the catalytic site, while other GPXs, including GPX5, GPX7 and GPX8, contain a cysteine residue. GPX1 is expressed in the lungs, kidneys, liver, and erythrocytes [80]. GPX2 is localized in the gastrointestinal-specific tissue; it is also present in other tissues, such as lung, liver, and skin. GPX3 is a glycosylated enzyme secreted in the plasma and extracellular fluid; its enzymatic activity is commonly used to evaluate the levels of Se in the organism. Strong activity of GPX4 is deserved in the testes. GPX4 has been found in mitochondria, cytoplasmic, and nuclear cellular compartments [81]. This enzyme is particularly active during cellular differentiation in embryonic development and in spermatogenesis [5]. Finally, GPX6 is found only in embryonic tissues. The three iodothyronine deiodinases (DIOs) are integral membrane proteins; DIO1 and DIO3 are plasma membrane proteins [82], while DIO2 is localized in the endoplasmic reticulum (ER) membrane. All these three DIOs participate in oxidoreduction enzymatic reactions with SeC residue in the catalytic site. These three enzymes are active in thyroid hormone metabolism by activating (DIO1 and DIO2) or inactivating (DIO3) tetraiodotyronine (T4), triiodothyronine (T3), and reverse-triiodothyronine (rT3) (Figure 2). Three DIOs exhibit different tissue distributions; indeed, DIO1 is present in the thyroid, liver, kidneys, and pituitary gland, while DIO2 is expressed in the thyroid, central nervous system, and skeletal muscle. Finally, DIO3 is present in the embryonic and neonatal tissues, uterus, and central nervous system. These thyroid hormones regulate also lipid metabolism, thermogenesis, and growth [83]. Three thioredoxin reductase (TrxR1, TrxR2, and TrxR3) are known, they are selenoproteins with the SeC at the penultimate position at the C-terminal end of the enzymes [84]. TrxR1 and TrxR2 are found in the cytoplasm and mitochondria, respectively, while TrxR3 is localized in testes [85]. Each monomer of these enzymes includes a FAD prosthetic group, an NADPH binding site, and a redox active disulfide site. During the reaction, the electrons are transferred from NADPH and FAD to the disulfide active site of TrxR, which reduces the various substrates as in Figure 3. TrxR reduces oxidized thioredoxin that provides a reducing equivalent to the disulfide bond of thioredoxin reductase, ribonucleotide reductase, thioredoxin peroxidase, and some transcription factors acting as a cell growth factor in DNA synthesis [5]. Many other substrates have been identified for this reductase, such as lipoic acid, lipid hydroperoxides, Ca-binding proteins, and glutaredoxin 2 [5]. Selenophosphate synthetase 2 (SPS2) catalyzes the synthesis of selenophosphate (Figure 1) in the presence of ATP, which transfer a phosphoryl group to hydrogen selenide (HSe^–^). Selenoprotein P is a very abundant glycoprotein in plasma and is formed by two domains: the C-terminal very rich in selenocysteine (10 residues) and the N-terminal, larger than the C-terminal. It is also present in the liver, brain, and testes [86]. The high level of selenocysteine suggests that it can play an important role in selenium transport and storage in tissues [87]. Selenoprotein P has antioxidant properties and helps to eliminate peroxynitrite formed in the reaction of superoxide and nitric oxide.

## 7. Selenium Deficiencies in Food: Keshan and Kashin–Beck Diseases

Keshan disease is a juvenile cardiomyopathy with pulmonary edema caused by a combination of both a nutritional deficiency of the essential mineral Se and a mutated strain of the Coxsackie B virus [88,89]. The name of the disease derives from Keshan County of Heilongjiang province, Northeast China, where the symptoms were first noted: precordial oppression and pain, nausea and vomiting (yellowish fluid), and myocardial necrotic lesions. Afterwards, these symptoms were found prevalently in a region extending from northeast to southwest China, where the soil is deficient in Se. Keshan disease may lead to cancer, hypertension, strokes, and, in addition, eczema, psoriasis, arthritis, and cataracts. Supplementations with Se reduce these conditions. In regions in which Se is present in low amounts, human beings can increase their intake of selenium with food, such as Brazil nuts, onions, canned tuna, beef, cod, turkey, chicken breast, eggs, cottage cheese, oatmeal, white or brown rice, and garlic. Moreover, human beings must be advised to have a diet rich in selenium that includes seafood and meats (kidney and liver). Onions, mushrooms, broccoli, tomatoes, and radishes are good sources of selenium if the soil, in which they are cultivated, contains it. Moreover, vitamin E deficiency is considered to have a relationship with the occurrence of Keshan disease, thus it is recommended to take vitamin E along with Se. Kashin-Beck disease is also due to the deficiency of Se and involves children between 5 and 15 years old living in areas with low Se levels. Kashin-Beck disease is a chronic osteochondropathy (disease of bones), mainly distributed from Northeastern to Southwestern China; other affected areas are Southeast Siberia and North Corea. Se and iodine have been considered the major nutritional deficiencies of this disease [90,91]. Furthermore, other causes of Kashin-Beck disease include mycotoxins trichotecene, produced by fungi (*Alternaria* sp. and *Fusarium* sp.), that contaminate barley grain in the diet, and fulvic acid in drinking water. Morning stiffness joints, limited motion in many joints of the body, disturbances of flexion and extension of the elbow, and enlarged interphalangical joints can be included among the symptoms of this disease. There are several Se supplements for treating Kashin-Beck disease in children, such as sodium selenite, sodium selenite, and vitamin E, sodium selenite and vitamin C, selenium salts, and selenium-enriched yeast. All these types of supplementations are highly effective compared to placebo/no treatment in treating Kashin-Beck disease in children. However, the content of Se must be strictly controlled to prevent harmful health effects, since Se in high doses may be poisoning [92].

## 8. Selenium Epigenetics: DNA, Histones, and Micrornas

There has been increased concern surrounding exposure to heavy metals (Cd, Cr, Hg, Fe, Ni, Pb) due to the evolving understanding of their role in the development of cancer [93,94,95]. Furthermore, the interaction between metalloids, such as Se and As, and the human body is correlated with the risk of lung, liver, urinary tract, kidney, and prostate cancers [96,97,98,99]. The essential trace element Se influences gene expression via different epigenetic pathways. Epigenetics refers to alterations in gene expression without involving any change in the DNA sequence but modifying the structural organization of chromatin. The mechanisms that mediate epigenetic regulation of gene expression are DNA hypo- or hyper-methylation, post-translational modifications of histone tail, and small non-coding RNA molecules (microRNA, miRNA). Eukaryotic DNA is packaged in the form of chromatin with the nucleosomes as a basic repeating unit. Each nucleosome is formed from 147 DNA base pairs wrapped around histones H2A, H2B, H3, and H4, which aggregate each other forming the histone octamer. The N-terminal end of these histones may undergo a variety of post-translational modifications promoted by specific enzymes with covalent reactions of acetylation and deacetylation, methylation and demethylation, phosphorylation, citrullination, sumoylation, biotinylation and ubiquitination processes. These reactions influence the chromatin structure facilitating gene transcription or its inhibition. DNA methyltransferases, histone methyltransferases, histone acetyltransferases, and histone deacetylases are the enzymes implicated in epigenetic mechanisms [94,95,97,100]. Dietary Se affects covalent reactions of DNA methylation and demethylation as well as histone methylation and demethylation, acetylation, and deacetylation. Se exerts its effects via the methionine-homocysteine cycle (one-carbon cycle) [4] (Figure 4). In this cycle, methionine is converted, in the presence of ATP and methionine adenosyltransferase (MAT), to S-adenosyl methionine (SAM), which is utilized as a methyl donor for methylation reactions of cytosine fifth carbon to 5-methylcytosine in DNA, and Se to dimethylselenide and dimethyldiselenide. During these methylation processes, SAM is transformed to S-adenosylhomocysteine which, in turn, is converted to homocysteine in the presence of the enzyme hydrolase (HY). Moreover, in this cycle, the methylation process of homocysteine to methionine takes place [101]. DNA methylation status is dependent on dietary intake of folate and vitamin B12, involved in the one-carbon cycle. The covalent methylation of DNA, in the presence of DNA methyltransferase and SAM, involves the transfer of a methyl group to the cytosine forming 5-methylcytosine; moreover, 5-methylcytosine can be actively or passively demethylated. Arai and coauthors [102] treated murine embryonic stem cells with Se supplementation at concentrations found in human maternal serum. These authors found that Se induced a reversible alteration of the cell heterochromatin status and decreased the DNA methylation level in the Aebp2 gene (a component of the epigenetic regulator polycomb repression complex) and Piekle2gene (related to neural differentiation) without acting on the cell potential to form embryonic bodies. To test if Se affects the reaction of DNA methylation and the gene regulation, Uthus et al. [103] use a methylation array in the human epithelial Caco-2 cells (immortalized cell line of colorectal adenocarcinoma cells). DNA from cells grown with 250 nM methylselenocysteine solutions was incubated with methylation-binding protein labeled with biotin and then hybridized to the methylation array. DNA genes with methylated promoters will produce higher chemiluminescence than those genes without a methylated promoter. The methylation reaction of the von Lippel–Lindau tumor gene suppressor, among the genes profiled, was decreased, and that this results in the downregulation of this tumor suppressor. de Miranda and collaborators [104] in their study demonstrate that methylselenic acid (MSA) and selenite are promoting anti-breast cancer, acting in a dose-dependent manner on MCF-7 human breast adenocarcinoma cells, and showing modulation of DNA methylation and histone post-translation covalent modification. MSA (2 μM) and selenite (8 μM) altered epigenetic marks involving decreased expression of DNMT1 and histone modification. In particular, MSA decreased trimethylation reaction on H3K9 (H3K9me3) and increased the acetylation reaction of H4K16 (H4K16ac), while selenite acted decreasing the H4K16ac histone mark [104]. The H3 and H4 histone tails undergo post-translational covalent reactions that include methylation, acetylation, phosphorylation, ubiquitination, and sumoylation. The acetylation of the N-terminal lysine residue is a major histone modification involved in transcription, chromatin structure, and DNA repair. Histones acetylation and deacetylation are reactions allowed by histone acetyltransferases and histone deacetylases (HDAC). Histones are also methylated on arginine and histidine residues. Methylation, unlike acetylation, does not alter the charge of the histone molecules. In the acetylation reaction, in the presence of acetyl coenzyme A, an acetyl group is transferred on the –NH^3+^ of the N-terminal lysine residue of histones that loses its ability to bind the DNA-negative phosphate group. This process releases negative charges of the phosphate groups of DNA, destabilizing the compact DNA-histone structure, and leading to a relaxed chromatin structure. Lee and collaborators [105] found that organoselenium compounds, such as methylselenocysteine and selenomethionine (Table 2), are metabolized to HDAC inhibitors in human prostate cancer cells (LNCaP cells). The selenocompounds methylselenocysteine and selenomethionine undergo transamination reaction in the presence of the enzymes glutamine transaminase K and aminoacid oxidase producing β-methylselenopyruvate and α-keto-γ-methylselenobutyrate, that are structurally similar to butyrate, a known HDAC inhibitor. These selenoketoacids decreased HDAC activity and increased the level of histone H3 acetylation in LNCaP cells. In a paper by Narayan and coauthors it has been shown, that in the presence of selenite and selenomethionine, the acetylation reaction of histone H4 (H4-K5, H4-K8, H4-K12, and H4-K16) was decreased in murine RAW 264.7 cells, macrophages and COX-2 promoter [106]. Finally, small non-coding microRNAs (miRNAs) are single-stranded, and are formed by 25–30 nucleotides. They play a central role in cell differentiation and proliferation and are involved in the post-transcriptional regulation of proteins expression binding to complementary target messenger RNA (mRNA) and silencing their translation, also reducing expression through induced decapping and deadenylation [107,108]. miRNAs are subjected to a covalent methylation modification (6-methyladenosine; 6MeA) and demethylation coordinated by methyltransferases and demethylases. Maciel-Dominguez and coauthors [109] studied the effect of Se on the gene expression of miRNA. Caco-2 cells were grown in Se-deficient or Se-supplemented medium [109]. After 72 h, RNA was extracted and subjected to microarray analysis (737 microRNA). Among 145 miRNA expressed in Caco-2 cells, 12 miRNA showed altered expression after Se-depletion. In the same study, the authors found that the expression of 50 mRNA was altered after Se depletion, and several mRNAs were targeted by the Se-responsive miRNA. MicroRNA-185, whose expression was silenced after Se-depletion, was confirmed to regulate the expression of glutathione peroxidase 2 (GPX-2) and selenophosphate synthetase 2 (SPS-2). SPS-2 contributes to selenoprotein biosynthesis machinery. In a recent study, Matouskova et al. predicted several miRNAs as putative regulators of many glutathione peroxidases [110]. Moreover, miRNA-185 is a target of Se, since it has been found as a tumor suppressor in ovarian, breast, prostate, and gastric cancer [111,112]. In hepatocarcinoma cell lines, Se treatment modulated miRNA-544a expression interacting with selenoprotein K [113]. Xiang et al., treating LNCaP (Limph Node Carcinoma of the Prostate, androgen-sensitive human prostate adenocarcinoma) cells with 1.5 μM selenite for one week, found an important reduced HDAC activity and, at the same time, an increase of H3-K49 acetylation [114]. Telomeres are sequences at both the end of eukaryotic chromosomes. Human telomeres end is formed of about 100 base pairs of repeated sequences (TTAGGG)n; in mammals, n is generally ten of thousand. The end of the linear chromosomes is not replicated by DNA polymerase, but a special mechanism, in the presence of the enzyme telomerase, adds telomeres to chromosome ends. In the absence of this mechanism, telomeres could be shortened progressively with cell divisions. In any case, the telomere length is an important strategy to prolong life and to act against free radicals and aging. In 2004 Liu et al. showed that sodium selenite (2.5 μmol/L) prolongs life and anti-aging enhancing the activity of telomerase and length of telomere of human hepatocytes L-02 [115]. Furthermore, sodium selenite at a relatively high level (10 μmol/kg) increased the activity and the expression of telomerase in rat hepatocites [116].

## 9. Phytoremediation and Rhizofiltration

Anthropogenic activities (industrialization, mining, cement plants, and burning coal to produce energy) and natural activities (seepages and weathering from rocks, volcanic activity, and forest fires) are the major causes of the presence of toxic metals in the environment [117]. Toxic heavy metals are not degradable and their accumulation in the soil and water can contaminate drinking water, v and fruits with deleterious effects on human health. Physical, chemical, and biochemical approaches for heavy metals removal from water and soil are expensive and invasive and do not provide solutions to this problem. It is not yet clear, whether Se is essential and important for vegetation growth; however, plants are able to absorb and accumulate Se in roots and leaves. The capacity of plants to uptake Se is utilized to remove this element from contaminated soils and water [94,95,97]. This process is termed phytoremediation and its popularity is increasing as a low-cost and friendly technology for remediating contaminated environments (soils, groundwater and surface water). Among different phytoremediation strategies, the processes of phytovolatilization, phytoextract,tion and rizhofiltration are well known. With the process of phytovolatilization, green plants are able to absorb inorganic Se and release it into the atmosphere as volatile selenocompounds, which are less toxic than original inorganic forms. It has been demonstrated that the volatile Se compound dimethyldiselenide (DMDSe, Table 2) is released by Astragalus bisulcatus and *Astragalus Racemosus*, while broccoli (*Brassica oleracea*) release dimethylselenide (DMSe, Table 2) from their leaves [118]. These organic selenocompounds are almost 600 times less toxic than elemental Se. The capability of different plant species in volatilizing selenium has been reported, showing that *A. bisulcatus* and *B. oleracea* have the highest rates of volatilization followed by tomatoes (*Solanum lycopersicum*), tall fescue (*Festuca arundinacea*) and alfalfa (*Medicago sativa*). Broccoli and cabbages (*B. oleracea)* are capable of volatilize daily at a Se quantity of 10g/ha [119,120]. In plants, Se detoxification takes place by methylation of selenocysteine and selenomethionine (Table 2) in the presence of the enzyme selenocysteine methyltransferase and S-adenosyl methionine (SAM) [121,122]. Among different crops grown on selenite silty soil, Indian mustard (*Brassica juncea*), corn (*Zea mays*), rice (*Oriza sativa*), and wheat (*Triticum aestivum*) proved a good tolerance to Se in the phytoextraction process. The Se accumulation in this plant was 104.8 mg/kg in Indian mustard, 76.9 mg/kg in rice, and 18.9 mg/kg in wheat [123]. The highest accumulation of Se for *Heliantus annuus*, *Gaillardia aristata*, *Calendula officinalis*, *Tagetes erecta*, *Coreopis gladiate*, and *Helichrysum orientale* make these flowers potentially attractive for phytoextraction from seleniferous soils [124]. The great advantage obtained by using flowers in the phytoextraction process derives from the consideration that they do not become part of the food chain of humans and animals. Rhizofiltration is another remediation process of absorption and concentration of contaminants from aquatic environments into the roots and shoots of both aquatic and land plants [125,126]. For the making of this method, plants are grown on the contaminated site (in situ) or they are grown off-site and later introduced to the contaminated aquatic ecosystems environment (ex situ). Rhizofiltration can treat the agricultural runoff, industrial discharge, and acid mine drainage released as a result of anthropogenic activities. Contaminants are absorbed through plant roots until saturation is reached, and finally, the plants are harvested with their roots. The plant species such as iris-leaved rush (*Juncus xiphioides*), cattail (*Typha latifolia*), hydrilla (*Hydrilla verticillata*), dotted duckweed (*Landoltia punctata*), common reed (*Phragmites australis*) and saltmarsh bulrush (*Scirpus robustus*) have given great results for Se rhizofiltration in wetlands.

## 10. Conclusions

Se is considered an essential trace element of fundamental importance for human health and its multifaceted aspects have attracted worldwide clinical and research interest in the last few decades. It is very important to maintain an adequate level of Se since both deficiency and excess are dangerous for human health. However, the crucial factor is that this micronutrient has a narrow range of safety. Indeed, while additional Se intake may benefit people with low status, those of adequate-to-high status may be affected adversely and should not take selenium supplements. Se performs its biological activity mainly through selenoproteins which are responsible for thyroid hormone management, fertility, the aging process, and immunity, and which play a key role in the maintenance of a redox balance in cells. Several studies have shown the interference of Se with epigenetic marks that are associated with the risk of diseases, or predict them. This is a relatively new field of the investigation thus, a detailed and comprehensive understanding of epigenetic processes elicited by Se is required to clarify and predict its impact on health outcomes. Se has been shown to act as anticarcinogenic in experimental settings and also in some human studies; however further extensive research in this field is necessary to determine the exact involved mechanisms. In this context, uncovering the putative roles of miRNAs as mediators of Se-dependent tumor protection against malignant transformation could be an interesting area of future studies.

## Figures and Tables

**Figure 1 ijms-24-02633-f001:**
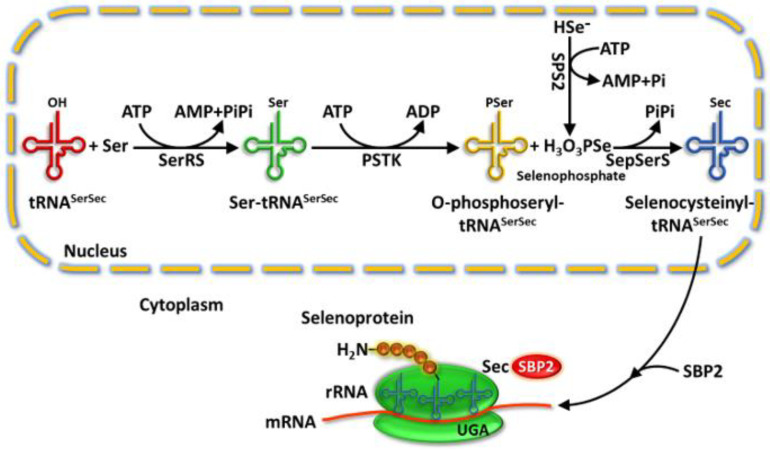
Selenoprotein biosynthesis pathway. SerRS catalyzes the reaction between tRNA^SerSec^ and Ser in the presence of ATP to yield Ser-tRNA^SerSec^, which in turn is phosphorylated in the presence of ATP and PSTK to give O-phosphoseryl- tRNA^SerSec^. Thus, O-phosphoseryl- tRNA^SerSec^ is replaced by Se in the presence of selenophosphate and SepSerS to synthesize Sec-tRNA^SerSec^. Sec-tRNA^SerSec^ is transferred to the ribosome thanks to SECIS and SBP2. Finally, the UGA codon is recognized as the Sec integration codon into the amino acid sequence of selenoprotein. Abbreviations: SeC = Selenocysteine; SerRS = Seryl-tRNA synthetase; mRNA = messenger RNA; rRNA = ribosomal RNA; PSTK = O-phosphoseryl tRNA kinase; SPS2 = selenophosphate synthetase 2; SepSerS = O-phosphoseryl tRNA: selenocysteinyl tRNA synthase; SECIS = SeC Insertion Sequence; SBP2 = SECIS Binding Protein 2.

**Figure 2 ijms-24-02633-f002:**
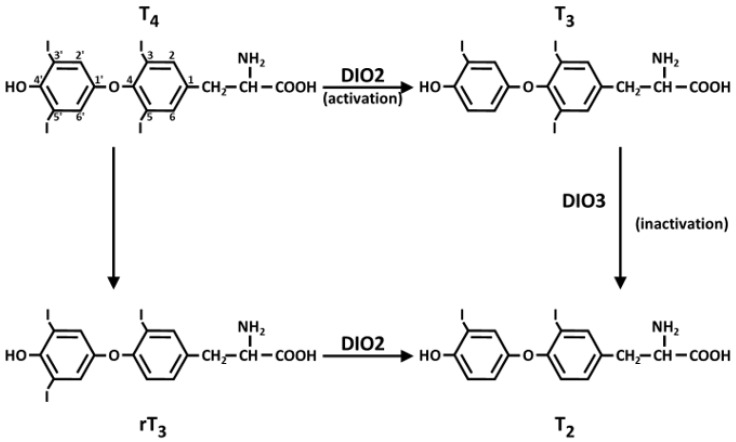
Deiodinases and metabolism of thyroid hormones. DIO2 deiodinase converts T4 into T3. DIO3 deiodinase mediates inner-ring deiodination of T4 or T3 to form the inactive metabolites rT3 and T2, respectively. Abbreviations. T4: thyroxine or 3,3′,5,5′-tetraiodotyronine; T3: 3,3′,5-triiodotyronine; rT3: reverse-3,3′,5′-triiodotyronine; T2: 3,3′-diiodotyronine.

**Figure 3 ijms-24-02633-f003:**
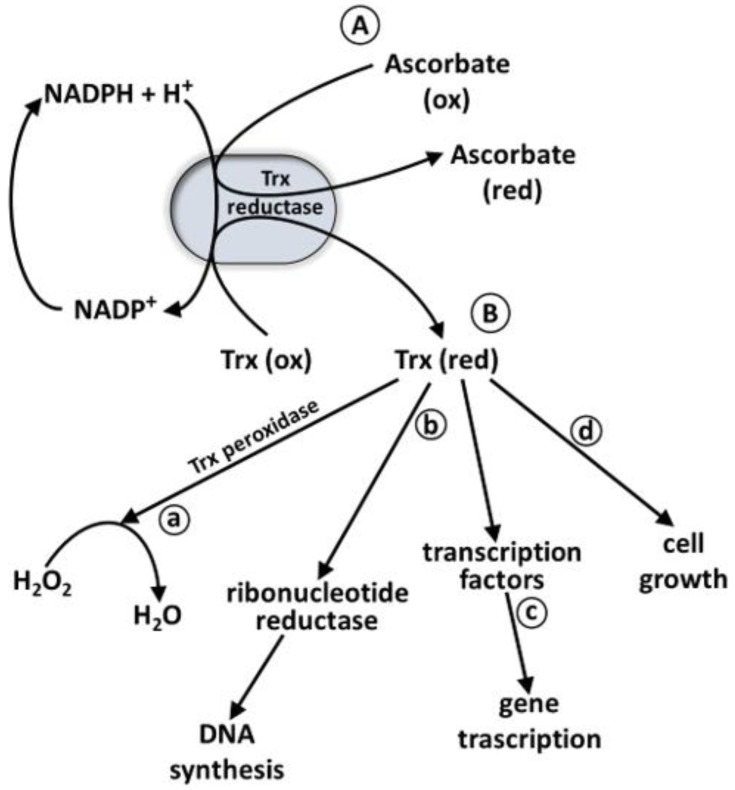
Reactions and functions of TrxR. (**A**) NADPH + H^+^ catalyzed in the presence of TrxR the reaction of Trx (Ox) into Trx(Red) to reduce ascorbate (Ox) into ascorbate (Red). (**B**) Reduced Trx supplies reducing equivalents to break down H_2_O_2_ to H_2_O in the presence of Trx peroxidase (**a**); reduced Trx in the presence of ribonucleotides reductase reduces ribonucleotides to deoxiribonucleotides for DNA synthesis (**b**); Trx (Red) provides reducing equivalent for transcription factors resulting in their increased binding to DNA leading to altered gene transcription (**c**); reduced Trx increase cell growth (**d**). Abbreviations: NADP^+^: Nicotinamide Adenine Dinucleotide Phosphate; NADPH: Nicotinamide Adenine Dinucleotide Phosphate (reduced form); Trx: thioredoxin; Trx (Ox): thioredoxin oxidized; Trx (Red): thioredoxin reduced; TrxR: thioredoxin reductase.

**Figure 4 ijms-24-02633-f004:**
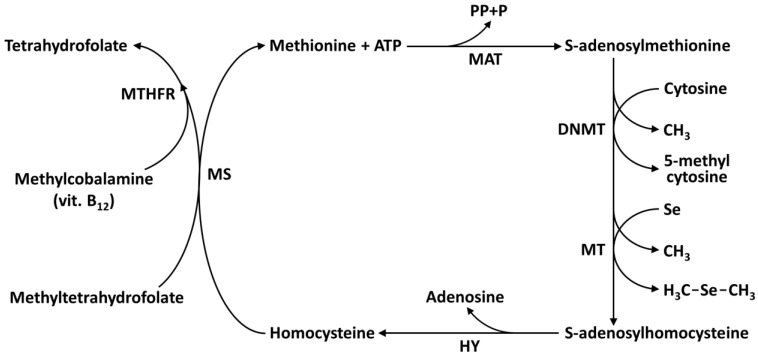
One-carbon metabolism cycle: DNA methylation and Se metabolism. Abbreviations: MAT: Methionine Adenosyl Transferase; MS: Methionine Synthase; MT: Methyl transferase; DNMT: DNA Methyl Transferase; HY: Hydrolase; MTHFR: Methylene Tetra Hydro Folate Reductase; PP: Pyrophosphate; P: phosphate.

**Table 1 ijms-24-02633-t001:** Chemical and physical properties of selenium.

Atomic number Atomic weight Electronic configuration Melting point Boiling point Density at 20 °C Covalent radius Van der Waals radius Heat of fusion (gray) Heat of vaporization Pauling electronegativity number First ionization energy Second ionization energy Third ionization energy Fourth ionization energy Standard potential Allotropes Mohs hardness Cristal structure (gray) Oxidation states	34 78.96 u [Ar] 3d^10^4s^2^4p^4^ 221 °C 685 °C 4.81 g/cm^3^ 120 ± 4 pm 190 pm 6.69 KJ/mol 95.48 KJ/mol 2.55 941.0 KJ/mol 2045.0 KJ/mol 2973.0 KJ/mol 4144.0 KJ/mol 0.823 V (VI/IV) Gray, Red, Black 2.0 Hexagonal −2, 0, 2, 4, 6

**Table 2 ijms-24-02633-t002:** Organic forms of selenium.

Molecular Structure	Name
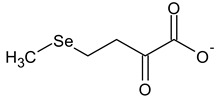	α-Keto-γ-methylselenobutyrate
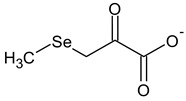	β-Methylselenopyruvate
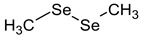	Dimethyldiselenide
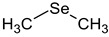	Dimethylselenide
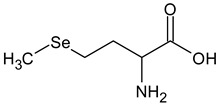	Selenomethionine
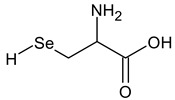	Selenocysteine
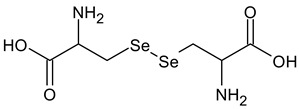	Selenocystine
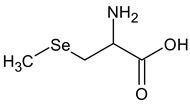	Methylselenocysteine
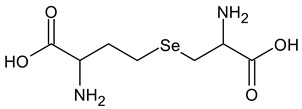	Selenocystathionine

**Table 3 ijms-24-02633-t003:** Selenium content in food products according to reference [1].

Food	Selenium Content (µg/g)	References
Yeast	500–4000	[18,58,59]
Brazil Nuts	0.2–512	[60]
Beef Kidney	1.45	[61]
Liver	0.3–0.4	[62]
Beef	0.01–0.73	[59,63]
Fish	0.06–0.63	[59,61,64,65]
Bread	0.09–0.20	[64,66]
Eggs	0.09–0.25	[59,63,67]
Chicken	0.15	[64]
Pork	0.27–0.35	[64,66]
Broccoli	0.012	[63]
Milk	0.01–0.06	[64,65]
Chocolate	0.04	[59,61,63,64,65]

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
