# Peer review of "Biological Activity of Selenium and Its Impact on Human Health"

_ijms, 2023, doi:10.3390/ijms24032633_

Round 1

Reviewer 1 Report

In this manuscript, the authors have briefly described in detail the role of biological role of Se and its function during the normal homeostasis of the human body and in case of its deficiency, how does the metabolism disturb which ultimately leads to impaired physiological functions. Overall, the manuscript is interesting and fits in the merits for publication in this journal but subject to the following concerns:

1.     Why is it so important to discuss the chemical and physical properties of Se in table 1. What is the role of these properties in the biological activities of Se and how do these properties exhibit their impact on the human health? In stead to this, it would be better if the authors include a section in which they describe in detail the occurrence of Se in the ecosystem and functional foods.

2.     Figure 2 looks very impressive, but it lacks its legend or caption. The interactions and interconversion of various biomolecules should be briefly described in the figure legend.

3.     What is the correlation between Selenoproteins and selenoenzymes. How do these two perform their functions. Do these biomolecules perform their functions without each other or not?

4.     It would be better if the authors also discuss in details the role of Se in metabolic disorders and their homeostasis .

5.     There are several grammatical mistakes and syntax errors. The whole manuscript needs critical revision to remove all the grammatical mistakes and syntax errors.

Author Response

  1. 01. 2023

Dear reviewer,

enclosed you will find the revised version of our manuscript “Biological Activity of Selenium and its Impact on Human Health” by Giuseppe Genchi, Graziantonio Lauria, Alessia Catalano, Maria Stefania Sinicropi, Alessia Carocci. We carefully revised our review trying to address your suggestions. Changes made have been evidenced in the text with “Track Changes” function.

  1. A table reporting the occurrence of Se in food products has been added (see Table 3)
  2. The interactions and interconversion of various biomolecules have been briefly described in the Figure 1 (previously reported as Figure 2) legend.
  3. From publications in the literature, about 50 selenoproteins (i.e. proteins that contain selenocysteine) are known, of which 25 are present in humans. Five glutathione peroxidases, three iodothyronine deiodinases, three thioredoxin reductases, seleno-phosphate synthetase 2 and methionine sulfoxide reductase B1 show activity as biological catalysts, i.e. they are enzymes. The other 12 selenoproteins show no enzymatic activity but are directly or indirectly involved in a myriad of physiological and metabolic pathways. Selenoproteins and selenoenzymes perform their functions independently of each other.
  4. The role of Se in metabolic disorders has been reported in the paragraph “Selenium in human health and diseases”.
  5. The whole manuscript has been checked in order to remove grammatical and syntax errors.

We are grateful to you for your constructive criticism that allowed us to improve our manuscript and we hope that the changes introduced in the version that we are now submitting may be considered sufficient to make the quality of our paper satisfactory.

Best regards

Maria Stefania Sinicropi

Reviewer 2 Report

Journal   IJMS (ISSN 1422-0067)

Manuscript ID         ijms-2182260

Type         Review

Title    Biological Activity of Selenium and its Impact on Human Health

Dear authors:

This review presents some aspects of the biological activity of selenium and its impact on human health.

In some countries, recommendations have been developed and programs have been adopted for the fortification of foodstuffs (eg bread) with selenium.

The manuscript presents interesting research with an in-depth review of the scientific literature since 1957.

The title of the review is good and specific enough to keep the reader interested, so it's easy to understand.

The topic of the review is quite interesting and relevant at the present time. The co-authors tried to analyze the accumulated experience in studying some properties of selenium.

The review is quite logically structured and complies with the principles of presenting scientific information and research.

The structure of the review corresponds to the purpose of the study.

I think there are enough tables and illustrations in the Review. Tables and figures are clear.

Please clarify - Are the copyrights of the drawings respected?

I believe that the authors of the manuscript approached the analysis of the problem quite carefully and used a number of citations (103).

To date, more than 10,000 publications devoted to the study of selenium are presented in various scientific databases.

The co-authors of the review took it upon themselves to use the necessary number of sources of information in accordance with their structure of the manuscript and the plan of their own research.

Introduction

In order to improve the presented overview, I recommend adding a clearly stated purpose of the study in this section.

Conclusions

The co-authors of the review recommend further study of the properties of selenium.

Less than 25 percent (24.04%) of the sources of information presented in the Directory (bibliographic list) have been published over the past 5 years.

References

13. Institute of Medicine. Dietary Reference Intakes for Vitamin C, Vitamin E, Selenium, and Carotenoids; The National Academies Press: Washington, DC, USA, 2000; p. 528.

20. Dhillon, K.S.; Bañuelos, G.S. Overview and prospects of selenium phytoremediation approaches. In: PilonSmits E, Winkel L, Lin ZQ (eds) Selenium in plants. Plant Ecophysiology, 2017, vol 11. Springer, Cham.

29. Langner, B. E. Selenium and selenium compounds. Ullmann's Encyclopedia of Industrial Chemistry, 2000

85. Mohr, A. M.; Mott, J.L. Overview of microRNA biology. In Seminars in liver disease (2015 Vol. 35, No.01, pp. 003–011). Thieme Medical Publishers

You need clarification on the design accepted in your journal

I recommend for publication the Review "Biological Activity of Selenium and its Impact on Human Health" - ijms-2182260.

Thanks for giving me the opportunity to read your work. 

Prof. Dr. Maksim Rebezov

V. M. Gorbatov Federal Research Center for Food Systems of Russian Academy of Sciences, Moscow, Russian Federation

  Максим Борисович Ребезов            17.01.2023

E-mail: rebezov@ya.ru

Author Response

Dear reviewer,

enclosed you will find the revised version of our manuscript “Biological Activity of Selenium and its Impact on Human Health” by Giuseppe Genchi, Graziantonio Lauria, Alessia Catalano, Maria Stefania Sinicropi, Alessia Carocci. We carefully revised our review trying to address your suggestions. Changes made have been evidenced in the text with “Track Changes” function.

  1. In the introduction, a sentence assessed the purpose of the review has been added.
  2. Further references published over the last 5 years have been added.
  3. References have been corrected according to referee suggestions.

We are grateful to the referees for their constructive criticism that allowed us to improve our manuscript and we hope that the changes introduced in the version that we are now submitting may be considered sufficient to make the quality of our paper satisfactory.

Best regards

Maria Stefania Sinicropi